# Impact on Population Health of Baltic Shipping Emissions

**DOI:** 10.3390/ijerph16111954

**Published:** 2019-06-01

**Authors:** Lars Barregard, Peter Molnàr, Jan Eiof Jonson, Leo Stockfelt

**Affiliations:** 1Unit of Occupational and Environmental Medicine, Department of Public Health and Community Medicine, Institute of Medicine, Sahlgrenska Academy, University of Gothenburg & Sahlgrenska University Hospital, Gothenburg, SE 405 30, Sweden; peter.molnar@amm.gu.se (P.M.); leo.stockfelt@amm.gu.se (L.S.); 2Norwegian Meteorological Institute, Oslo, NO 0313, Norway; janeij@met.no

**Keywords:** air pollution, shipping, Baltic Sea, SECA, health effects, mortality, ischemic heart disease, stroke

## Abstract

Emission of pollutants from shipping contributes to ambient air pollution. Our aim was to estimate exposure to particulate air pollution (PM_2.5_) and health effects from shipping in countries around the Baltic Sea, as well as effects of the sulfur regulations for fuels enforced in 2015 by the Baltic Sulfur Emission Control Area (SECA). Yearly PM_2.5_ emissions, from ship activity data and emission inventories in 2014 and 2016, were estimated. Concentrations and population exposure (0.1° × 0.1°) of PM_2.5_ were estimated from a chemical transport mode, meteorology, and population density. Excess mortality and morbidity were estimated using established exposure-response (ER) functions. Estimated mean PM_2.5_ per inhabitant from Baltic shipping was 0.22 µg/m^3^ in 2014 in ten countries, highest in Denmark (0.57 µg/m^3^). For the ER function with the steepest slope, the number of estimated extra premature deaths was 3413 in total, highest in Germany and lowest in Norway. It decreased by about 35% in 2016 (after SECA), a reduction of >1000 cases. In addition, 1500 non-fatal cases of ischemic heart disease and 1500 non-fatal cases of stroke in 2014 caused by Baltic shipping emissions were reduced by the same extent in 2016. In conclusion, PM_2.5_ emissions from Baltic shipping, and resulting health impacts decreased substantially after the SECA regulations in 2015.

## 1. Introduction

International shipping is one of the sources of air pollution. Sulfur oxides, nitrogen oxides, and particulate matter (PM) are emitted from ship smokestacks, and these emissions have global effects on health and the environment. The International Maritime Organization (IMO) has proposed a new global standard to limit sulfur (S) in fuel oil from 2020 from the current limit of 3.5% to 0.5% sulfur. Both the North Sea and the Baltic Sea are defined by IMO (International Maritime Organization) as SECAs (Sulfur Emission Control Areas). SECAs are sea areas in which stricter controls were established to minimize airborne emissions from ships as defined by Annex VI of the 1997 MARPOL Protocol [1]. A more detailed description of SECA can be found in a paper by Cullinane and Bergqvist [2].

The EU sulfur directive requires ships to use fuel with 0.1% sulfur in harbor areas from January 2010. Further reductions to 0.1% are mandatory in SECAs from January 2015. Prior to July 2010, the maximum allowed sulfur content in SECAs was 1.5%. Further global reductions of fuel sulfur are also planned [3]. Fuel sulfur reduction will have a significant impact on emitted SO_2_, as well as PM, since SO_2_ is a precursor for PM [4].

Population-weighted exposure to air pollution from shipping has been estimated globally based on emission inventories and global scale atmospheric models [5,6,7]. Exposure data were combined with regional mortality rates and exposure-response functions for mortality. The estimated number of annual premature deaths in the latest estimate was about 350,000 [7]. A scenario using a reduction of fuel sulfur to 0.5% globally was estimated to decrease premature mortality, due to shipping emissions by 34% [7]. 

Estimates for Europe were also performed by Andersson et al. [8] and Brandt et al. [9]. In both studies, international shipping was estimated to cause about 50,000 annual premature deaths in Europe. Shipping in the Baltic Sea and the North Sea was estimated to cause about 14,000 annual premature deaths in Europe in 2011, with about 6% decrease in 2020 after reductions of fuel sulfur.

These global and European estimates were performed with a relatively low resolution (0.1 – 1° × 0.1–1° grids) and mortality rates were estimated on a regional level. In addition, some country-specific estimates have been performed. For Denmark, Brandt et al. estimated that international shipping caused about 500 premature deaths in 2011, 400 of which were due to shipping in the Baltic Sea and the North Sea, with a decrease of 13% in 2020 [10].

There is lack of detailed and updated regional data in areas affected by the SECA regulations, such as the Baltic Sea, and no estimates of effects on mortality of “real life” changes of emissions after the application of the SECA regulations.

The aim of the present study was to perform detailed estimates of exposure to particulate air pollution from shipping in the Baltic Sea in countries bordering the Baltic Sea, and assess the long-term effects on mortality and morbidity from such exposure. Moreover, the possible health effects of the sulfur regulations for marine fuels enforced in January 2015 in the Baltic SECA area were estimated.

## 2. Materials and Methods 

The calculations of the air pollutants, including PM_2.5_, focusing on the Baltic Sea region are described by Jonson et al. [11], and only a short summary is given here. Ship emissions have been calculated with the STEAM model and are based on the actual ship movements from AIS (Automatic Identification System) calculated by the STEAM model [12,13]. These data consist of hundreds of millions of automatic position reports sent by ships with an automatic transponder system. Combined with the characteristics of each ship and engine type, the emissions from each individual ship were calculated. In the Baltic Sea ship emissions from 2016 and 2014 (before and after the implementation of stricter SECA regulations in 2015) are used. For all other sea areas, 2015 emissions are used. As the emissions are used for multiple years the ship emissions of NOx, SOx, CO, and PM are aggregated to monthly values. The 2016 land-based emissions are from IIASA-Eclipse [14].

The concentrations of air pollutants have been calculated with the EMEP model [15,16] with later model updates described in Simpson et al. [17] and references therein with a 0.1 x 0.1 degrees model resolution. Through regular model validation with measurements and model inter-comparisons the performance of the EMEP model is well documented, see references in Jonson et al. [11], which also describes an additional model validation, including measurements close to the Baltic Sea. 

In addition to a base run, including all emissions, two model sensitivity runs are made. In the first model sensitivity run, all Baltic Sea emissions are excluded. The difference between the base run and the first sensitivity run represent the contribution from Baltic shipping to ambient air pollution. In the second sensitivity run, the 2016 Baltic Sea emissions are replaced by 2014 (high sulfur) emissions representing the decrease in air pollutant concentrations following the implementation of a stricter SECA in 2015. All model scenarios have been made for the three meteorological years 2014, 2015 and 2016 in order to cancel out meteorological variability.

Gridded population (1 ×1 km) density by country was obtained from Eurostat for 2011 [18] and used to calculate population-weighted exposure to PM_2.5_ (in µg/m^3^ × number of persons) from Baltic shipping based on output from the EMEP modelling. The data were extrapolated to 2015 with population sizes for that year. Gridded population data with a similar resolution was not available for Russia. Instead, for 72 million people, residing in relevant parts of European Russia, population-weighted exposure was estimated from Administrative Unit Center Points from NASA SEDAC (Socioeconomic Data and Applications Center) in 2010 [19]. Population-weighted exposure from Baltic shipping was also averaged by country.

Age-specific mortality rates for 2015 were obtained from Eurostat [20]. For the present study, we used listed total mortality from ≥25 years of age, which represents 98–99% of total mortality. Natural mortality was approximated as 95% of total mortality, which is the typical fraction for Northern Europe.

We used two alternative exposure-response (ER) functions for natural mortality based on long-term effects of particulate air pollution. The first one was the WHO HRAPIE recommendation regarding concentration-response functions related to air pollutants for the metrics (“Group A”) for which enough data were considered available to enable quantification of effects [21]. For annual mean PM_2.5_, a relative risk of 1.0062 (95% CI 1.004–1.008) per µg/m^3^ is recommended for natural mortality. Recommendations are given also for some specific causes of death (e.g., lung cancer), and for mortality related to daily mean PM_2.5_, but these outcomes are included in the relative risk for natural mortality. WHO HRAPIE also suggests some concentrations-response functions for hospital admissions related to daily mean PM_2.5_. The disease burden is, however, dominated by natural mortality as a long-term effect.

The second ER function was based on the large European multi-center ESCAPE project [22]. The confounder-adjusted relative risk (hazard ratio) was 1.014 per µg/m^3^ (95% CI 1.004–1.026). This was based on a meta-analysis of 19 cohorts from 13 countries, among them Sweden, Finland, Denmark, and Germany. 

The population attributable fraction of disease (PAF; in this case natural mortality) was calculated from the relative risk (RR) at the specific exposure level as (RR−1)/((RR-1) + 1). The PAF was then applied to the background of natural mortality per country to calculate the extra mortality attributed to air pollution from shipping.

The years of life lost (YLL) were estimated from life tables obtained from the national statistics units and Eurostat assuming increased mortality from Baltic shipping as indicated above. 

For estimates of morbidity, we used data on baseline incidence of ischemic heart disease (IHD) and stroke from the Global Burden of Disease (GBD) database [23], and exposure response-functions from the ESCAPE-study for acute coronary events [24] and stroke [25]. The relative risks were 1.026 (95% CI 1.00–1.06) per µg/m^3^ of annual mean PM_2.5_ for IHD and 1.038 (95% CI 0.98–1.12) per µg/m^3^ for stroke. To avoid double counting with mortality estimates we subtracted the numbers of deaths, due to ischemic heart disease and stroke from the incidence, again using the GBD database and assuming that half the deaths were from new (incident) cases of IHD/stroke.

## 3. Results

The EMEP model results used in this study are documented in Jonson et al. [11] for relevant pollutants. Only the results for PM_2.5_ are included here. The estimated emissions of from Baltic shipping in 2014 contributed up to 20% of total PM_2.5_ levels in some coastal areas in Denmark, Sweden, and Finland, while for most parts of Northern Europe the contribution was <1% [11]. After the implementation of the SECA regulations emissions and air pollution from shipping decreased substantially (Figure 1).

The contribution of Baltic shipping to population exposure depends on the relationship between population density and air pollution levels. Population exposures for PM_2.5_ in ten European countries in 2014 and 2016, using meteorology for 2014–2016 are shown in Table 1 and vary considerably between meteorological years. The mean exposure was highest in Denmark (about 0.5 µg/m^3^ PM_2.5_), followed by Sweden, Estonia, Finland, Latvia, and Lithuania, while the highest total population exposure (in µg/m^3^ × persons) was highest in Germany and Poland, due to their large populations (Table 1, Figure 2). There was a clear reduction in population exposure, due to decreasing emissions from 2014 to 2016. Using the mean meteorology of 2014–2016, the reduction was 34%. The contribution of Baltic shipping to population exposure was about 10% of total levels of PM_2.5_ in coastal areas of the Baltic SECA area, but <1% in remote areas (Figure 2).

Total natural mortality and estimated number of premature deaths, due to PM_2.5_ emissions from Baltic shipping in 2014 and 2016, are shown in Table 2 using the two alternative ER functions mentioned above. The numbers were largest in Germany and Poland in line with their large populations. The number of estimated premature deaths, due to Baltic shipping, decreased from about 1500 in 2014 to about 1000 in 2016. The reduction was on average 37% and was highest in the countries neighboring the Gulf of Finland.

The number of years of life lost decreased from about 17,000–38,000 with 2014 emissions to about 11,000–T25,000 with 2016 emissions. The number of YLL per premature death varied slightly between countries, due to differences in age-specific death rates. 

Estimated morbidity from non-fatal IHD and stroke, due to PM_2.5_ emissions, from Baltic shipping in 2014 and 2016 is shown in Table 3. The numbers of extra cases were highest in Germany and Poland, mainly due to large exposed populations. They were also relatively high in Denmark and Sweden, due to higher exposures to PM_2.5_ from shipping, and in Russia, due to a large population. The number of incident cases of IHD and stroke, due to PM from Baltic shipping, decreased by around 500 each after 2015, roughly a decrease by one third.

The estimated numbers for premature deaths, YLL, IHD, and stroke presented for 2014 and 2016 in Table 2 and Table 3 are annual numbers, so these numbers are (approximately) valid also for the years before 2014, and after 2016, respectively. 

## 4. Discussion

The present study suggests that the stricter SECA regulations on fuel sulfur for the Baltic have indeed been successful in reducing adverse health effects, due to air pollution from shipping, reducing its impact on mortality, YLL, and morbidity by at least one third. A positive impact of SECA on health was predicted in some previous forecasts [6,9], but this is the first study using actual empirical data on emissions before (2014) and after (2016) implementation of the SECA regulations. The calculated decrease in emissions was based on the STEAM model [12,13]. Many studies have demonstrated large reductions of shipping emissions after fuel sulfur reductions. On example is a study following a ship that switched from high-sulfur fuel to low-sulfur fuel when entering regulated waters of California [26]. 

Our results showed a larger impact on mortality after the decrease in fuel sulfur than predicted by Winebrake et al. [6] and Brandt et al. [10]. However, the number of YLL in Denmark, due to Baltic shipping, in the present study when the HRAPIE ER function was used (1900 in 2014 and 1400 in 2016) are consistent with the estimates by Brandt et al. [10], who used the same ER function and estimated about 4100 YLL pre-SECA (in 2011) and 3600 post-SECA (in 2020). Brandt et al. considered not only the Baltic Sea, but also shipping emissions in the North Sea. The study by Winebrake et al. [6] used a much lower resolution (1 × 1 degree) for air pollution modelling, while the grid size used by Brandt et al. [10] for air pollution and population density was consistent with the present study.

Emissions were calculated assuming 100% compliance with the SECA regulations after 1 January 2015. Separate studies have indicated that compliance is indeed high, varying from 89 to 99% in fairways examined [27].

Stricter regulations on marine fuel sulfur could possibly cause a model shift towards land-based transports [28]. In the case of the Baltic Sea, this seems, however, not to have been the case [29]. 

Even though the present study had a grid size of 0.1 × 0.1 degrees for air pollution modelling, this resolution will probably underestimate population exposure to shipping pollution and thereby adverse health effects. The reason for this is that population density usually is higher very close to the coastline where the contribution of air pollution from shipping is highest. The extent of such underestimation of exposure can be evaluated using high resolution air pollution modelling.

The exposure-response functions for PM_2.5_ versus mortality used in the present study were based on two recent sources. The HRAPIE review was based on a meta-analysis of 11 epidemiological studies by Hoek et al. [30] and has been widely used in health impact assessments [21]. On the other hand, the ESCAPE study included European countries, several of them (Sweden, Finland, Denmark, and Germany) bordering the Baltic Sea [22]. The ER function from the ESCAPE study is about twice as steep as the one reported in the HRAPIE review. This may be due to the fact that the ESCAPE studies were based on within-city estimates, while the studies used in the HRAPIE review also included between-city estimates. A recent meta-regression using estimates from 53 different studies found a relative risk of 1.013 per µg/m^3^ of PM_2.5_ when air concentrations were around 10 µg/m^3^ of PM_2.5_ [31]. Therefore, we consider our results based on the ER function from the ESCAPE study (Table 2) somewhat more likely than the estimate based on the HRAPIE review.

The estimates for both mortality and morbidity were based on studies of long-term exposure mainly using annual exposures to PM. Long-term studies of PM air pollution generally find higher risk estimates than short-term studies, but using ten years of exposure rather than one only increases the risk marginally more. This indicates that the majority of the cardiovascular health effects are due to relatively rapid biological responses, such as increased thrombotic potential, endothelial dysfunction or plaque instability—and that these effects are potentially reversible within one or a few years after an intervention that reduces PM exposure [32]. 

Population exposure to total PM_2.5_ from shipping is a mixture of primary PM compounds emitted (elemental carbon, organic and inorganic PM) and secondary inorganic particles (SIA; sulfates, nitrates, and ammonium) produced by chemical reactions over hours and days. It has been estimated that in the long range transported PM reaching populated areas, SIA accounts for about 80% of total PM exposure [8]. There are some indications that primary combustion PM has stronger effects on mortality than SIA and some researchers, therefore, applied different ER functions to assumed fractions of total PM [8]. However, in the large U.S. studies, SIA constituted a major fraction of total PM, and showed the same ER function as the one used in the HRAPIE review [33]. In that study, associations between mortality and sulfate particles were consistent with associations between mortality and total PM. We chose to apply a single ER function to the total PM_2.5_ contribution from shipping in line with the health impact assessment by Brandt et al. [9] and Sofiev et al. [7]. The exposure-responses for PM_2.5_ and IHD and stroke were selected from the ESCAPE study, since it includes multiple cohorts from the relevant countries. 

Obviously, the contributions from Baltic shipping to total PM exposure are highest in coastal areas (Figure 2). In these areas, Baltic shipping contributes about 10% of the total adverse health effects from particulate air pollution, which is not negligible. Nevertheless, also minor contributions to PM levels contribute to the health impact if populations are large (Table 2). In the 2000, global shipping was estimated to contribute 7.4% of total PM_2.5_ exposure in Europe with the highest contributions in the Mediterranean [34]. A reduction of PM exposure by limiting fuel sulfur would, therefore, save many European lives and avoid many cases of heart disease and stroke, as well as other diseases that are not included in this assessment. 

This is the first study using empirical data on emissions and meteorology to model the effects on air pollution of the SECA regulations and estimate the beneficial health effects of lowering fuel sulfur. Another strength is the use of relatively detailed modeling of air pollution and estimates of populations. As mentioned above, a limitation of the study is the fact that high resolution (less than 1 × 1 km) data on air pollution and populations were not available. Moreover, the three years modelled (2014–2016) may not have captured all variability in meteorology. There is also some uncertainty regarding the exposure-response functions used, since they are based on air pollution contrasts, due to mixtures of emissions, often dominated by road traffic. 

## 5. Conclusions

The present study of health impacts from shipping in the Baltic Sea indicates that the SECA regulations on lower fuel sulfur have had substantial effects on population exposure to PM_2.5_ in coastal areas, thereby reducing premature deaths, ischemic hearts disease, stroke and years of life lost from shipping emissions. For example, the number of estimated extra premature deaths decreased by about one third in 2016 (after SECA), a reduction of >1000 cases. This is an example of how environmental policy development on air pollution can directly improve the health of the population.

## Figures and Tables

**Figure 1 ijerph-16-01954-f001:**
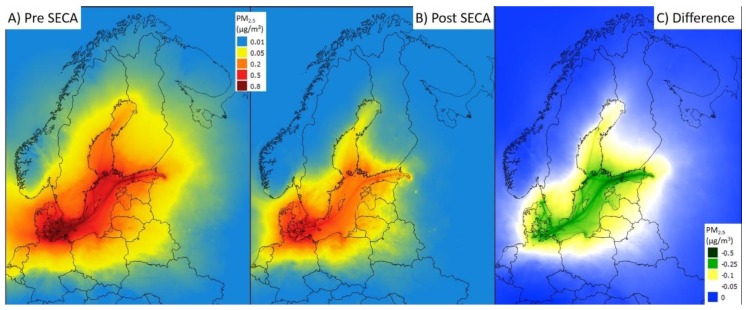
Estimated contribution of shipping emissions in the Baltic Sea to PM_2.5_. (**A**) in 2014 (before the SECA regulations of marine fuel sulfur). (**B**) in 2016 (after the SECA regulations of marine fuel sulfur). (**C**) Difference between 2014 and 2016. In all cases, the estimated contributions to PM_2.5_ were modelled using the same meteorology (average for 2014–2016).

**Figure 2 ijerph-16-01954-f002:**
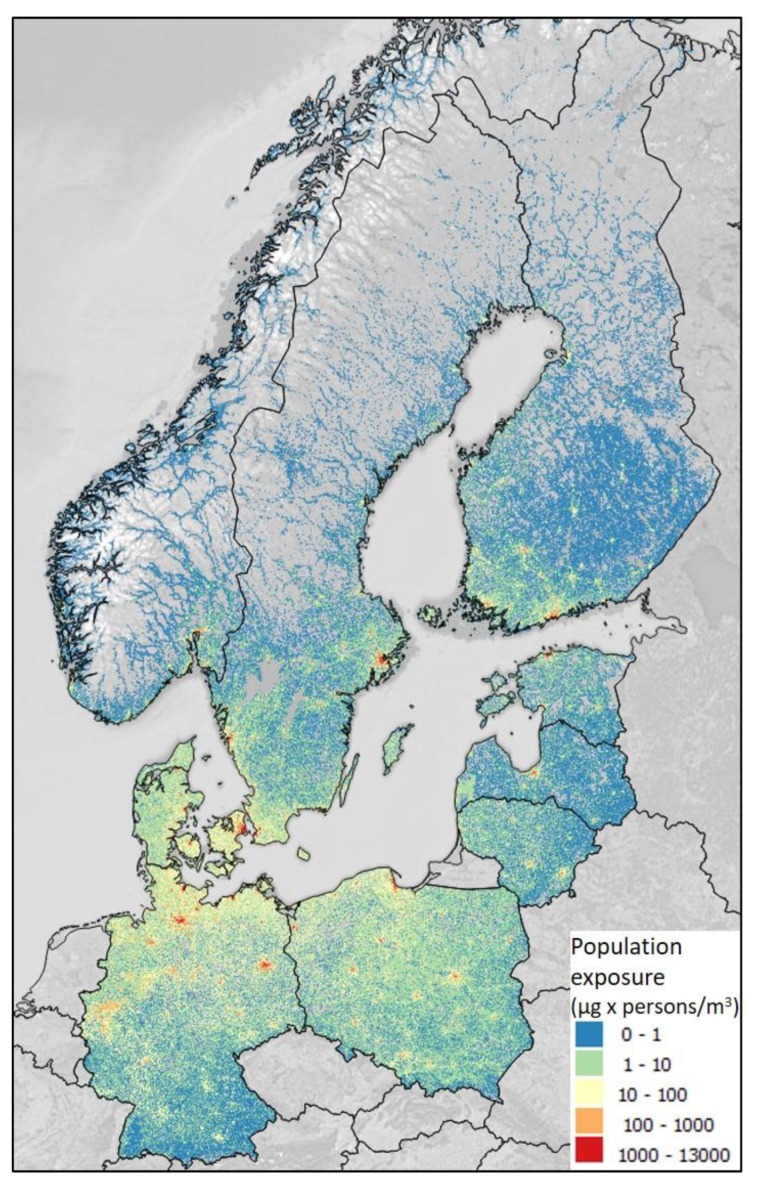
Estimated contribution of shipping emissions in the Baltic Sea to population exposure of PM_2.5_ in 2016 (after the SECA regulations of marine fuel sulfur) in each square of 0.1^o^ × 0.1^o^ (about 10 × 10 km). The unit is µg/m^3^ PM_2.5_ × number of persons.

**Table 1 ijerph-16-01954-t001:** Population exposure of PM_2.5_ (in µg/m^3^ × number of persons × 106) and mean exposure per inhabitant from contributions of Baltic shipping to ambient PM_2.5_ concentrations in ten European countries. The table shows emissions (“E”) in 2014 (before SECA) and 2016 (after SECA) under meteorological conditions (“M”) in 2014, 2015, and 2016. The numbers for Russia only considers the European part of Russia.

Country	Population 2015 × 10^3^	Population Exposure E 2014, M 2014	Population Exposure E 2014, M 2015	Population Exposure E 2014, M 2016	Population Exposure E 2014, M mean	Mean Exposure per Person	Population Exposure E 2016, M 2014	Population Exposure E 2016, M 2015	Population Exposure E 2016, M 2016	Population Exposure E 2016, M Mean	Mean Exposure per Person
Sweden	9747	3630	2962	3531	3374	0.354	2344	1918	2273	2178	0.228
Norway	5166	267	205	299	257	0.052	158	125	188	157	0.032
Denmark	5660	3430	2534	3440	3135	0.566	2669	1888	2522	2360	0.426
Finland	5472	1288	1200	1327	1272	0.238	697	643	694	678	0.127
Germany	81,198	7100	5414	8505	7006	0.087	5174	3865	6248	5096	0.064
Poland	38,006	3119	4048	4720	3962	0.103	2031	2757	3198	2662	0.069
Estonia	1315	413	431	481	441	0.341	229	238	261	243	0.188
Latvia	1986	359	525	507	464	0.223	214	326	296	278	0.134
Lithuania	2921	455	656	638	583	0.193	300	423	400	375	0.124
Russia	72,450	2784	3499	3197	3160	0.044	1526	1896	1615	1679	0.024
Sum, Mean	223,921	22,844	21,475	26,645	23,655	0.220	15,342	14,080	17,696	15,706	0.142

**Table 2 ijerph-16-01954-t002:** Estimated number of premature deaths (natural mortality), due to PM_2.5_ emissions from Baltic shipping in 2014 and 2016 according to two alternative exposure response functions. The left one of the two numbers given refers to the ER function suggested in the HRAPIE report [21] and the right one the ER function found in the ESCAPE study [22].

Country	Mortality at Age > 25 in 2015 (*n*/Year)	Premature Deaths per Year in 2014	Years of Life Lost in 2014	Premature Deaths per Year in 2016	Years of Life Lost in 2016	Reduction (%)
Sweden	90,103	187–421	1812–4092	120–272	1167–2635	35
Norway	40,312	12–28	127–287	8–17	78–176	39
Denmark	52,111	173–390	1901–4293	130–294	1431–3231	25
Finland	53,536	75–169	775–1750	40–90	414–935	47
Germany	919,548	471–1063	4940–11,155	342–773	3634–8206	27
Poland	390,815	236–532	2868–6476	158–358	1922–4340	33
Estonia	15,121	30–68	346–781	17–38	191–431	45
Latvia	28,237	37–83	414–935	22–50	249–562	40
Lithuania	41,339	47–105	514–1161	30–68	330–745	36
Russia ^a^	958,514	245–553	2977–6722	134–302	1625–3670	45
Sum, Mean	2,621,754	1511–3413	16,674–37,651	1001–2261	11,041–24,932	37

^a^ Only including the European part of Russia, closer to the Baltic Sea.

**Table 3 ijerph-16-01954-t003:** Estimated number of extra annual cases of ischemic heart disease and stroke, due to PM_2.5_ emissions from Baltic shipping in 2014 and 2016.

Country	Extra Cases of IHD 2014	Extra Cases of IHD 2016	Reduction (*n*)	Extra Cases of Stroke 2014	Extra Cases of Stroke 2016	Reduction (*n*)	Reduction (%)
Sweden	208	134	74	180	116	64	35%
Norway	13	8	5	18	11	7	39%
Denmark	210	158	52	169	127	42	25%
Finland	93	50	44	100	53	46	47%
Germany	521	379	142	465	338	127	27%
Poland	231	155	76	254	170	83	33%
Estonia	34	19	15	36	20	16	45%
Latvia	28	17	11	47	28	19	40%
Lithuania	44	29	16	58	37	21	36%
Russia ^a^	166	91	75	228	125	103	45%
Sum, Mean	1548	1039	510	1555	1026	528	37%

^a^ Only including the European part of Russia, closer to the Baltic Sea.

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
