# Peer review of "Impact on Population Health of Baltic Shipping Emissions"

_ijerph, 2019, doi:10.3390/ijerph16111954_

Round 1

Reviewer 1 Report

The paper deals with an interesting subject on impacts of shipping pollutants on health. While there have been some studies (notably Lack 2007), there has not been much research on this topic.

Please be consistent with the use of English. US English: sulfur, UK: sulphur. You are currently using both.

You should introduce the topic of SECAs a little bit more in the introduction as not all readers would be familiar with the regulation. You could use the following:

Cullinane, K., & Bergqvist, R. (2014). Emission control areas and their impact on maritime transport.

Also, you have to take into consideration that while the SECAs reduced emissions for harmful pollutants (mainly due to use of better fuel), there are some additional emissions that are generated because of the modal shifts to landbased modes. See the following paper on modal shifts:

Zis, T., & Psaraftis, H. N. (2017). The implications of the new sulphur limits on the European Ro-Ro sector. Transportation Research Part D: Transport and Environment, 52, 185-201.

Also, lines 211-218 you should try to update the values of the contriubtion of shipping to the different types of pollutants. Using data from 2000 is very old (long before even the first ECAs were designated)

It is not clear to me how you have come up with the pollution levels. Are you using any dispersion model? or simply estimate the generated pollutatns at each grid and based on that you calculate exposure? Because do this right, you need dispersion modelling as well. You also need to provide other impacts of SECAs. for example, if speeds went lower because of the regulation, the emissions intensity could increase for PM pollutants. At lower sailing speeds the combustion is sub optimal and the emission factor is increasing.

Finally, I would like to know (I am an engineer so I am not that familiar with the way premature deaths are linked) about your results in Tables 2 and 3. Is it possible to quantify if the difference in mortality rates are due to the lower sulphur fuel? Wouldn't it be the case that many of the deaths or strokes attributed to emissions would happen much later to the affected people? Because I would expect a reduction in mortality taking effect much later (e.g. people dying in 2016 could still die because of the longterm effects of pollution in earlier years).

Author Response

Reviewer 1

1. “The paper deals with an interesting subject on impacts of shipping pollutants on health. While there have been some studies (notably Lack 2007), there has not been much research on this topic. Please be consistent with the use of English. US English: sulfur, UK: sulphur. You are currently using both.”

Response

Thank you for this comment. Yes, we tried to cover the literature on impacts of shipping pollutants on health. Since the reviewer specifically referred to “Lack 2007”, we searched the literature for a paper by Daniel Lack in 2007 on impact on of shipping pollutants health effects. We found a number of papers by Daniel Lack on shipping emissions, although no paper on shipping emissions and health from 2007. However, we found the paper by Lack et al. from 2011, interesting for our discussion on impact of fuel sulfur on shipping emissions, and added a reference to that study in the Discussion section.

We also agree that the spelling of sulfur must be consistent and have changed “sulphur” to “sulfur”.

2. “You should introduce the topic of SECAs a little bit more in the introduction as not all readers would be familiar with the regulation. You could use the following: Cullinane, K., & Bergqvist, R. (2014). Emission control areas and their impact on maritime transport.”

Response

We agree, and have added the following text in the Introduction: “SECAs are sea areas in which stricter controls were established to minimize airborne emissions from ships as defined by Annex VI of the 1997 MARPOL Protocol [6]. A more detailed description of SECA can be found in Cullinane and Bergqvist (2014) [7].”

3. “Also, you have to take into consideration that while the SECAs reduced emissions for harmful pollutants (mainly due to use of better fuel), there are some additional emissions that are generated because of the modal shifts to landbased modes. See the following paper on modal shifts: Zis, T., & Psaraftis, H. N. (2017). The implications of the new sulphur limits on the European Ro-Ro sector. Transportation Research Part D: Transport and Environment, 52, 185-201.”

Response

Yes, this is an important comment, and we added a para in the Discussion section on this and the reference. However, in this specific case, according to HELCOM (Baltic Marine Environment Protection Commission - Helsinki Commission) [HELCOM (Baltic Marine Environment Protection Commission - Helsinki Commission) http://www.helcom.fi/] there has been no reduction in sea traffic between 2014 (pre SECA) and 2016 (post SECA) and therefore the modal shift towards land-based transport due to stricter emission regulations seems not to be important. One reason that there was no reduction in sea traffic might be that global oil prices fell during the period when the SECA was introduced, thus negating the extra cost for cleaner fuel.

4. “Also, lines 211-218 you should try to update the values of the contribution of shipping to the different types of pollutants. Using data from 2000 is very old (long before even the first ECAs were designated).”

Response

We agree that the estimates from year 2000 (the reference Griffiths et al. 2011) are a bit old. Several papers have more updated maps of shipping contributions (so contributions by area), but we found no study in the literature with a more updated population exposure (contributions taking population density into account).  

5. “It is not clear to me how you have come up with the pollution levels. Are you using any dispersion model? or simply estimate the generated pollutants at each grid and based on that you calculate exposure? Because do this right, you need dispersion modelling as well. You also need to provide other impacts of SECAs. for example, if speeds went lower because of the regulation, the emissions intensity could increase for PM pollutants. At lower sailing speeds the combustion is sub optimal and the emission factor is increasing.”

Response

Yes, a dispersion model has been used. We have revised the previous text describing the air pollution modelling (the first three paragraphs under “2. Materials and Methods” in the revised version). Regarding the reviewer’s concern about speed, this is taken into account by the Helcom Automatic Identification System (AIS) in the STEAM model, and our contacts with ship owners suggest that the change of fuel sulfur did not cause any change of speed.

6. “Finally, I would like to know (I am an engineer so I am not that familiar with the way premature deaths are linked) about your results in Tables 2 and 3. Is it possible to quantify if the difference in mortality rates are due to the lower sulphur fuel? Wouldn't it be the case that many of the deaths or strokes attributed to emissions would happen much later to the affected people? Because I would expect a reduction in mortality taking effect much later (e.g. people dying in 2016 could still die because of the longterm effects of pollution in earlier years).”

Response

We are here referring to the long-term effects of air pollution. There is also a so called short-term effects, an increase in mortality on a single day with a high level of air pollution.

There are two aspects of the issue you raise. First, the long-term effects are expected to occur every year after a change in air pollution level and therefore the numbers given in Table 2 are “extra premature deaths per year” (after SECA 2016, 2017, 2018 etc.). We have now clarified this by stating that were are referring to long-term effects in the 6th para of the Introduction, and in the 4th para of the Materials and Methods section. We also added a para at the end of the Results section, clarifying that the estimated numbers for mortality and morbidity refer not only to the specific years 2014 and 2016, but for the years before 2015 and the years after 2015. Second, one might expect that it would take a number of years before a decrease of PM would impact health. This is true for cancer, which has a long induction/latency time. But regarding mortality and morbidity other than cancer, the “long-term” effects on health has been found to occur already a year after change in PM exposure. We have added a para, with references, on this in the Discussion section.

Reviewer 2 Report

The manuscript ID: sustainability-504193 entitled  Impact on population health of Baltic shipping  emissions  is interesting for International Journal Environmental research and Public Health.

However, in order to meet completely the aims of the journal the authors should follow some observations/suggestions indicated here below.

First, the manuscript does not highlights clearly the usefulness of the study, and its results, for the users (academics and practitioners). Especially, the authors does not clarify the really contribute of the results to the existent literature, as well as they does not clearify the approach adopted.

The manuscript revolves around three key pillars, that is the maritime regulation about the environmental issues (see MARPOL including all Annex and other), the community health, and a region (that is Baltic).

Reading the manuscript, I would like to read 1. a regulatory framework  about the first pillar, that includes the environmental regulatory framework for the shipping and for the ports; 2. a literature review on the topic proposed by the authors including the academics and practioners perspective, showing also the managerial instruments currently adopted; 3. a section dedicated to the methodology. The current version is confused.

Besides, the authors should to better explain the link between shipping emissions for a specific period and the main effects on the community health or population health.

The current version of the manuscript is also missing in terms of a costructive discussion for the academics and practitioners. The section 4 is not very clear because it includes some sentences that the authors should include in other sections of the manuscript structure (see lines 219-221).

The conclusions section (see lines 230-234) is missing.

I suggest to the authors to review the manuscript following the observations here above in order to improve the current version of the topic proposed.

Finally, the manuscript is missing of environmental policy development.

Author Response

Reviewer 2

“The manuscript ID: sustainability-504193 entitled  Impact on population health of Baltic shipping  emissions  is interesting for International Journal Environmental research and Public Health. However, in order to meet completely the aims of the journal the authors should follow some observations/suggestions indicated here below.”

1. “First, the manuscript does not highlights clearly the usefulness of the study, and its results, for the users (academics and practitioners). Especially, the authors does not clarify the really contribute of the results to the existent literature, as well as they does not clearify the approach adopted. The manuscript revolves around three key pillars, that is the maritime regulation about the environmental issues (see MARPOL including all Annex and other), the community health, and a region (that is Baltic). Reading the manuscript, I would like to read 1. a regulatory framework  about the first pillar, that includes the environmental regulatory framework for the shipping and for the ports; 2. a literature review on the topic proposed by the authors including the academics and practioners perspective, showing also the managerial instruments currently adopted; 3. a section dedicated to the methodology. The current version is confused.”

Response

As requested by the reviewer we have now re-structured the Introduction, starting (1) with a para on the regulatory framework (“first pillar”), then (2) the literature on the public health impact of shipping emissions (“second pillar”). Then follows (3) the section Materials and Methods.

2. “Besides, the authors should to better explain the link between shipping emissions for a specific period and the main effects on the community health or population health.”

Response

Thank you. Yes, as indicated above (Response to item 1.6) we have now explained that a permanent decrease in air pollution levels will have an effect on long term mortality and morbidity, and that such an effect can be expected to appear already in the first year after improvement of air quality.

3. “The current version of the manuscript is also missing in terms of a costructive discussion for the academics and practitioners. The section 4 is not very clear because it includes some sentences that the authors should include in other sections of the manuscript structure (see lines 219-221).”

Response

We hope that the Discussion section, with the additions and revisions will be seen as constructive. We revised the sentence (previous lines 219 – 221) suggested by the reviewer.

4. “The conclusions section (see lines 230-234) is missing.”

Response

We expanded the Conclusions section.

5. ”I suggest to the authors to review the manuscript following the observations here above in order to improve the current version of the topic proposed. Finally, the manuscript is missing of environmental policy development.”

Response

The manuscript as a whole describes the effect of an environmental policy development (the SECA regulations). We now added a sentence about this in the Conclusions section.

Round 2

Reviewer 1 Report

The authors have addressed my comments

Reviewer 2 Report

Dear Authors,

your paper is well done, but is so far by my competencies.
From my point of view, the study could be interesting if it includes also the managerial perspective that is missing in the second version of the manuscript.
In my first revision, I provided you some observations and suggestions but you have not included any aspects in your version.
I don't feel like expressing anything else.

Therefore,  I think that the editors will be decide about the publication of the manuscript that anyway it is well.